# Vascular Extracellular Vesicles Indicate Severe Hepatopulmonary Syndrome in Cirrhosis

**DOI:** 10.3390/diagnostics13071272

**Published:** 2023-03-28

**Authors:** Sukriti Baweja, Anupama Kumari, Preeti Negi, Swati Thangariyal, P. Debishree Subudhi, Shivani Gautam, Ashmit Mittal, Chhagan Bihari

**Affiliations:** 1Department of Molecular and Cellular Medicine, Institute of Liver and Biliary Sciences, New Delhi 110070, India; 2Department of Pathology, Institute of Liver and Biliary Sciences, New Delhi 110070, India

**Keywords:** hepatopulmonary syndrome, liver cirrhosis, vascular remodeling, endothelium, extracellular vesicles, diagnosis

## Abstract

Background: Hepatopulmonary syndrome (HPS) is a pulmonary vasculature complication in the setting of liver disease that is characterized by pathological vasodilation resulting in arterial oxygenation defects. We investigated the role of extracellular vesicles (EV) in cirrhosis patients with HPS, as well as the functional effect of EV administration in a common bile duct ligation (CBDL) HPS mouse model. Methods: A total of 113 cirrhosis patients were studied: 42 (Gr. A) with HPS and 71 (Gr. B) without HPS, as well as 22 healthy controls. Plasma levels of EV associated with endothelial cells, epithelial cells, and hepatocytes were measured. The cytokine cargoes were estimated using ELISA. The effect of EV administered intranasally in the CBDL mouse model was investigated for its functional effect in vascular remodeling and inflammation. Results: We found endothelial cells (EC) associated EV (EC-EV) were elevated in cirrhosis patients with and without HPS (*p* < 0.001) than controls. EC-EV levels were higher in HPS patients (*p* = 0.004) than in those without HPS. The epithelial cell EVs were significantly high in cirrhosis patients than controls (*p* < 0.001) but no changes found in patients with HPS than without. There was a progressive increase in EC-EV levels from mild to severe intrapulmonary shunting in HPS patients (*p* = 0.02 mild vs. severe), and we were able to predict severe HPS with an AUROC of 0.85; *p* < 0.001. An inverse correlation of EC-EVs was found with hemoglobin (r = −0.24; *p* = 0.031) and PaO2 (r = 0.690; *p* = 0.01) and a direct correlation with MELD (r = 0.32; *p* = 0.014). Further, both TNF-α (*p* = 0.001) and IL-1β (*p* = 0.021) as cargo levels were significantly elevated inside the EVs of HPS patients than without HPS. Interestingly, upon administration of intranasal EVs, there was a significant decrease in Evans blue accumulation and lung wet–dry ratio (*p* = 0.042; 0.038). A significant reduction was also noticed in inflammation and cholestasis. Conclusion: High levels of plasma EC-EV levels were found in patients with HPS with elevated pro-inflammatory cytokine cargoes. EC-EVs were indicative of severe HPS condition. In the CBDL HPS model, we were able to prove the beneficial effects of improving vascular tone, inflammation, and liver pathogenesis.

## 1. Introduction

Hepatopulmonary syndrome (HPS) is a lung vascular disorder in the setting of disease, with a clinical triad of impaired arterial oxygenation, alterations in pulmonary microvasculature, and advanced liver disease condition [1]. HPS is prevalent in 15–20% of patients with advanced liver disease [2]. Those with HPS have a higher risk of mortality due to liver cirrhosis and of associated complications of portal hypertension [3]. Currently, there are neither effective therapy nor non-invasive diagnostic tools for measuring severity for the hepatopulmonary syndrome. The only management of and treatment for HPS is liver transplantation, which also carries a higher risk of mortality in HPS patients [4]. The pathogenesis of HPS involves microvascular dilatation in the pulmonary arterial circulation. The changes are probably due to decreased pre-capillary arteriolar vascular tone and may be associated with other mechanisms such as angiogenesis, remodeling, and vasculogenesis [5]. The hallmark feature in humans for vasodilatation is the overproduction of nitric oxide (NO). Exhaled NO levels have been observed in cirrhotic patients, with the presence of HPS as a measure of pulmonary defects. These levels were found to be raised but to return to normal levels after liver transplantation, because HPS is also resolved [6,7,8,9]. However, until now, the exact mechanisms and pathogenesis of raised levels of NO production with portal hypertension, which influences the frequency of severity of liver injury, remain unknown. The high production of endothelin-1 and tumor necrosis factor (TNF) in the lung vasculature have aided in the development of experimental HPS models [10]. Microvascular endothelial defects appear to be the result of raised nitric oxide production from endothelial cells and also of the upregulation of inducible nitric oxide synthase and activity in intravascular macrophages [10]. In HPS patients, there are no definitive physical symptoms presented in the clinic, except in some cases in which the presence of cyanosis, digital clubbing, spider nevi, and severe hypoxemia is observed, which are suggestive of the syndrome [1]. The majority of patients develop progressive disease over a period of time, and there is worsening gas exchange [11,12]. There has been no sudden improvement noticed in HPS patients [13]. The survival rate rapidly declines in patients with liver diseases if they develop HPS. Hence, by the time patients present with severe symptoms, it is too late for effective treatment. There is an urgent need for effective diagnostic or prognostic markers for the severity of the disease. Thus, HPS remains an important and unmet medical issue.

Over the past decade, extracellular vesicles have drawn attention for their therapeutic potential in liver diseases. Extracellular vesicles (EVs) are membrane-bound, cell-derived vesicles that are released by all living cells and are present in biological fluids. There are three major subtypes of EVs based on their biogenesis and size, with exosomes being the smallest (40–100 nm), followed by microvesicles (100–1000 nm) and apoptotic bodies (>1 um), which are the largest. EVs are present in almost all biological fluids, such as plasma, urine, ascitic fluid, bile, etc., which are also reflective of the disease condition. EVs can act as the cargo of many of bioactive molecules among cells, including messenger RNA, metabolites, proteins, lipids, and even cytokines, and they are effective cellular “communicosomes” [14]. Additionally, due to the biogenesis of microvesicles (MV), they are able to carry surface markers and cargo, which may reflect the cell of origin as well as the specific stress factors that induced their formation and release. Changes in the levels and composition of EVs in biological fluids may reflect the status of different pathologies in liver diseases [15]. Our group has previously also demonstrated that EVs associated with stem cells and hepatocytes can be utilized as early predictors of response, even at the baseline, to steroid therapy in severe alcoholic hepatitis [16].

The aim of the present study was to determine the various origins of circulating EVs in cirrhosis patients that are or are not causally associated with hepatopulmonary syndrome. We investigated the comprehensive landscape of endothelial-, epithelial-, and hepatocyte-origin EVs in cirrhosis patients with or without HPS. Further, in order to understand the functional significance of EVs in vivo, we evaluated the potential contribution of the administration of healthy EVs as healthy cargo in a CBDL mouse model and explored their potential in regulating vascular tone, inflammation, and liver regeneration.

## 2. Methods

Patients: Inpatient and outpatient cirrhosis patients with and without the presence of hepatopulmonary syndrome were enrolled at the Institute of Liver and Biliary Science (ILBS), New Delhi, India and were screened for inclusion. The present study was ethically approved by the institute (F.25/5/107/ILBS/AC/2016/11252/299), and patients aged between 18–64 years with HPS were recruited after their written informed consent was received. Pulse oximetry was performed on all patients, and patients having an oxygen saturation level lower than 90% were followed up with by ABG and saline contrast echocardiogram for the presence of HPS. Patients with cardiopulmonary disorder, hepatic encephalopathy, usage of antibiotics in the last month or any current use of exogenous nitrates, sepsis, or the presence of hepatocellular carcinoma or any other malignancy were excluded from the study. The total (*n* = 42) number of cirrhosis patients with HPS (*n* = 71) and without HPS were studied along with age- and gender-matched healthy controls (*n* = 22).

Extracellular vesicles (EVs) isolation and characterization: EVs were isolated using differential ultracentrifugation [16] as previously described, and quality was assessed with a nanoparticle-tracking assay for size and concentration and stored at −80 C until further processing.

Enumeration of extracellular vesicles: After being frozen, EVs were thawed at room temperature and further characterized using flow cytometry (FACS) as previously described by our group [16]. Briefly, the EVs were washed using cold phosphate-buffered saline (PBS) and resuspended in PBS, and the surface antibodies were labelled for EVs associated with endothelial cells (CD31+ VECadherin+ AnnexinV+), epithelial cells (Epcam+ AnnexinV+), and hepatocytes (ASGPRII+ AnnexinV+). The antibodies were incubated for 30 min in the dark, washed, and acquired. The threshold of the FACS were set on Annexin V+ particles. The EVs were acquired using a BD FACS verse instrument (Beckman Coulter, Brea, CA, USA) and analyzed using FCS express software (Denovo Software, Pasadena, CA, USA).

Levels of cytokines (IL-1β, TNFα, IL-10) inside extracellular vesicles as cargo: The thawed EVs were pulse sonicated thrice for 5 s each and then centrifuged for 5 min, supernatants were collected, and the levels of cytokines such as IL-1β, TNFα, and IL-10 were measured using the sandwich ELISA method according to the manufacturer’s protocol (Elabsciences, Wuhan, China). The detection was performed chromogenically at wavelengths of 405 nm, 459 nm, and 545 nm, respectively, and concentrations were calculated in pg/mL.

Common bile duct ligation (CBDL) as HPS mouse model: Common bile duct ligation (CBDL) is an experimental model of obstructive cholestasis-induced chronic liver injury that is close to human hepatopulmonary syndrome for obstructive cholestasis, liver fibrosis, and portal hypertension. This results in bile duct proliferation, raised liver enzymes, inflammation, and hepatic fibrosis at 3–4 weeks post-surgery in mouse models. The study was approved by the institute’s animal ethics committee under approval number IAEC/ILBS/17/01. C57BL/6J male mice of 10–12 weeks were used. All animal care was performed according to the guidelines of care and use of laboratory animals by the Committee for the Purpose of Control and Supervision of Experiments on Animals (CPCSEA).

### 2.1. Surgical Procedure

All the animal studies were conducted in aseptic conditions. The mice’s surgery was conducted under 1.5–2% isoflurane anesthesia. Two knots of non-resorbable sutures were administered to ligate the common bile duct (CBD). Two weeks post-surgery of CBDL, the mice were administered intranasally with either healthy EVs (isolated from healthy wild-type C57/BL6J mice lungs) with a concentration of 10^8^ everyday (*n* = 6) or normal saline (vehicle control) (*n* = 6) for the next 2 weeks. In order to target the lungs, this method is safe, convenient, and easy to perform. The EVs directly reach the blood stream through its vascular bed, thus providing quick onset action with 100% bioavailability, as this route avoids gastrointestinal destruction and hepatic metabolism [17]. Control mice were sham operated on in the same way by the separating the the bile duct but without ligation (*n* = 6). The total duration of post-surgery was 4 weeks, and then all the animals were euthanized, and required procedures were conducted.

### 2.2. Lung Vascular Permeability Assessment Using Evans Blue Dye

As described previously [18], lung permeability was measured by two methods, one of which was the extravasation of Evans blue dye in the lung, which was injected retro-orbitally, and the second of which was the estimation of the lung wet–dry weight ratio, which was determined to quantify the lung vascular leak. Left and right lobes from the same mice were used for Evans blue albumin extravasation, followed by complete drying in an oven at 60 °C overnight for the estimation of the lung wet–dry ratio.

### 2.3. Histology

Both the tissues of liver and lung samples were formalin-fixed dehydrated and then embedded in paraffin and stained for hematoxylin and eosin (H&E) (liver, lung) to assess the pathology of the disease condition.

## 3. Statistical Analysis

All the baseline clinical characteristics of the patients were represented as mean ± SD or median IQR. Data were analyzed using an ANOVA test or a Mann–Whitney test, and Pearson correlation was used for the correlation analysis. All statistical tests were two-tailed, and the significance level was set at *p* < 0.05. Statistical analyses were performed using SPSS (IBM SPSS, Armonk, NY, USA) or GraphPad prism version 7.0.

## 4. Results

### 4.1. Baseline Clinical Characteristics of Cirrhosis Patients with or without HPS

We enrolled patients in the study from both the outpatient clinic as well from among the inpatients admitted at the Institute of Liver and Biliary Sciences (ILBS), New Delhi. This study cohort included a total of *n* = 113 patients and *n* = 22 healthy volunteers. The mean age in the cirrhosis with HPS group was 50.6 ± 9.7 years, and in the group without HPS, it was 50.2 ± 10.7 years (*p* = ns). The majority of patients were men in both the groups. The majority of cirrhosis etiology was alcoholism, followed by chronic hepatitis and NAFLD. Most of the clinical parameters were comparable among groups, as detailed in Table 1.

### 4.2. Various Cells Associated with Extracellular Vesicles’ Levels in HPS Patients

EVs are reflective of cellular stress or early apoptosis. To understand the cells of origin of the EVs, which are under stress, we estimated the EVs in the plasma of both groups of patients and compared this with healthy volunteers. Representative images from NanoSight, depicting the concentration and size of isolated EVs from all three groups, are shown in Figure 1A. The HPS in the liver disease setting is a vascular remodeling. For the same reason, we measured the EV levels associated with endothelial cells (CD31+ VECadherin+ AnnexinV+; EC-EVs), as well as epithelial cell (Epcam+ AnnexinV+)- and hepatocyte (ASGPRII+ AnnexinV+)-associated EVs. Interestingly, we found endothelial cell (EC)-associated EVs were significantly elevated in cirrhosis patients with and without HPS than healthy controls (*p* < 0.001). They were further found to be higher in HPS patients than in those without HPS (*p* = 0.004) (Figure 1B). However, the epithelial- and hepatocyte-associated EVs also had significant elevation levels in cirrhosis patients than controls (*p* < 0.001) (Figure 1C,D), but levels were comparable among cirrhosis patients with and without HPS. Further, upon clinical correlation, we found a direct correlation of EC-EVs with PaO2 levels (r = 0.690, *p* = 0.01) and MELD (r = 0.301, *p* = 0.014) (Figure 1E) and indirect correlation with PaO2 levels (r = −0.690, *p* = 0.01) and hemoglobin (r = −0.248, *p* = 0.031) (Figure 1E).

### 4.3. Elevated Levels of Endothelial Extracellular Vesicles Predict Severe Shunting

Further, the HPS condition was sub-grouped on the basis of shunting, i.e., mild, moderate, and severe. The EC-EV levels progressively increased from mild to moderate and to severe intrapulmonary shunting. Interestingly, we found a significant increase in EC-EV levels in severe shunting versus in mild shunting (*p* < 0.00) (Figure 2A).

Further, we performed a ROC analysis with the estimation of corresponding area under the curve (AUC) to calculate the indicative potential of EC-EVs for severe versus mild shunting in the HPS condition, which was found to be significant (AUC, 0.85; 95% CI, 0.725–0.985; *p* < 0.001), with a sensitivity of 83.3% and a specificity of 78.2% (Figure 2B).

### 4.4. Extracellular Vesicle Cargo in HPS Patients

It has been shown in previous studies [10] that macrophages play a significant role in the overproduction of inflammatory cytokines in the HPS condition. We wanted to investigate the functional role of EVs in HPS patients as cargo in carrying inflammatory molecules. Interestingly, we found both TNF-α (>4 fold; *p* = 0.001) and IL-1β (>2 fold; *p* = 0.021) (Figure 3A,B) levels were elevated inside the total EVs of HPS patients in comparison to those without HPS. On the contrary, no changes were found in anti-inflammatory IL-10 (*p* = ns) cytokines in the EVs (Figure 3C). However, the healthy volunteers’ EVs carried lower levels of cytokines than those of cirrhosis patients (*p* < 0.001).

### 4.5. Lung Vascular Changes after EV Administration in CBDL Mouse Model

Further, to investigate the functional and therapeutic role of EVs in the HPS condition, we developed a mouse models close to HPS. This was a common bile duct (CBD) model, which was ligated with two knots of a non-resorbable suture. After 2 weeks of CBDL, the mice were injected intranasally with extracellular vesicles (1 × 10^8^ EV/μL) obtained from healthy WT mice lungs, which were administered intranasally into the CBDL mice (2 weeks post-surgery of CBDL) every day for 2 weeks. The presence of intrapulmonary shunting was evaluated using Evans blue dye by albumin accumulation in the lungs. The trans-endothelial albumin influx was significantly higher in CBDL mice than in the sham group (*p* < 0.001) (Figure 4A). The lung wet–dry ratio was also found to be higher in CBDL than in the sham group by 4 weeks post-surgery (*p* = 0.00) (Figure 4B) The vasodilation-associated genes and inflammatory molecules’ mRNA levels were also studied (Figure 4C). Interestingly, upon administration of intranasal EVs, there was a significant decrease versus in CBDL mice.

### 4.6. Liver Histological Changes after EV Infusion in CBDL Mouse Model

The CBDL is an experimental model that is close to human HPS for obstructive cholestasis, liver fibrosis, and portal hypertension in mice. Liver histology after CBDL showed bile duct proliferation and inflammation, whereas no morphological changes in sham controls were observed. Liver fibrosis increased from an F0-1METAVIR score at week 1 to F4 at week 4 after CBDL. Interestingly, a significant reduction in inflammation and cholestasis was noticed after EV administration versus in the CBDL vehicle (Figure 5). Gene regulation also showed a reduction in the caspase9 apoptotic marker, which was significantly reduced after EV administration versus in the CBDL (*p* = 0.02), whereas other genes associated with hepatocyte proliferation were comparable.

## 5. Discussion

Our study found that cirrhosis patients with HPS have significantly elevated levels of endothelial cell-associated extracellular vesicles, which are directly correlated with MELD and carry high levels of pro-inflammatory cytokines as cargo. EC-EVs were also able to indicate severe HPS in cirrhosis patients. However, in a CBDL HPS mouse model, the administration of healthy EVs protects vascular integrity, decreasing permeability and reducing systemic inflammation.

HPS is common in cirrhosis patients, especially those who are on waiting lists for liver transplants. It induces severe pulmonary complications with a high risk of mortality. However, in animal models, a complex interaction between pulmonary endothelial cells and monocytes and the respiratory epithelium, high-producing cytokines, or angiogenic growth factors is responsible for the changes in the alveolar microvasculature, which results in defects in arterial oxygenation [19,20]. However, in our study, we observed EVs associated with endothelial cells, and epithelial cells were increased, which reflects the death of the cell of origin or stress. EC-EVs were further increased in HPS patients and were hence correlated with severity in cirrhosis patients. One of the main findings of our study is that the concentration of EC-EVs is inversely correlated with PaO2 levels in cirrhosis patients, which may aid in the diagnosis of HPS. This needs further investigation. Further, EC-EV levels were able to indicate severe shunting in HPS. Vascular endothelial cells have multiple critical roles in maintaining vascular homeostasis, especially in cellular communication, through the delivery of metabolites and even oxygen to tissues [21]. One major function of endothelial cells includes optimizing gas exchange, reducing vascular tone or aiding in the modulation of vasoconstriction. ECs can release their EVs directly in circulation via the bloodstream, from which they are delivered to various sites [22]. Hence, various functions of ECs can be performed via EVs. Endothelium dysfunction is also directly involved in several pathological disturbances. HPS is a vascular disorder in which the increase in permeability can lead to vascular leakage, edema, imbalance between vasoconstrictors, and vasodilatations, which eventually increase the pro-inflammatory condition. However, stress enforces the release of EVs from ECs. It has been reported that there are elevated levels of endothelial microparticles in vascular diseases, and that there is endothelial dysfunction in renal failure, coronary heart disease, and metabolic disorders [22]. Such studies indicate that EC-EVs can play a significant role in the pathogenesis of various diseases that have endothelial dysfunction. Our findings evidence the fact that in the HPS condition in advanced liver disease patients, EC-EV levels increase and may carry various information to distant sites, which may lead to increased severity of the disease.

The plasmatic load of EVs carrying inflammatory cytokines, such as tumor necrosis factor alpha (TNFα), a pro-inflammatory cytokine secreted in response to inflammatory stimuli, including endotoxin exposure, is also increased in human and experimental cirrhosis and in parallel circulating endotoxin levels. It was also shown that TNFα inhibition actually reduces the development of the HPS condition in CBDL rat models [23]. A study from our group showed promising results of both pentoxifylline, which is known to reduce TNFα synthesis, and anti-TNFα antibodies, which appear to resolve HPS through the prevention or reduction of the development of hyperdynamic circulation [24]. In our study, we investigated the proinflammatory cytokines TNFα and IL-1β, which were found to be increased in the EVs of HPS patients versus those without HPS, whereas anti-inflammatory cytokine IL-10 levels were comparable in EVs. Meanwhile, it has been shown that IL-10 is an immune suppressive, and its expression depends upon pro-inflammatory signals. It is found to be low in the plasma of cirrhosis patients [25,26]. It can be speculated that cytokine cargoes in EVs aid in cellular communication, which induces further pathogenic damage and increases the severity of the disease.

Recently, the recognition and understanding that exosomes modulate the angiogenic process of endothelial cells has been expanding [27]. Exosomal miR-194 mediates the cross-talk between hepatocytes and endothelial cells and contributes to proliferation, migration, and tube formation. The exosome/miR-194 axis plays a critical pathologic role in pulmonary angiogenesis [27]. We also administered EVs intranasally into the CBDL mice every other day for 4 weeks. We also observed the improvement in lung wet–dry ratio and inflammatory gene expression. EVs are heterogenous in population. Further studies are required to investigate the status of EVs carrying different metabolites and the resolution of HPS with different cargoes, sources, and routes of EV administration.

In conclusion, this article is the first to describe high levels of plasma EC-EV concentrations in patients with HPS. EC-EVs are associated with intrapulmonary shunting and may resemble a new prognostic biomarker of early shunting. In the CBDL HPS model, we were also able to prove the beneficial effects of EC-EVs in improving vascular tone, inflammation, liver pathogenesis, and overall mortality rates. Hence, EC-EVs play a significant role in the HPS condition in advanced liver disease.

## Figures and Tables

**Figure 1 diagnostics-13-01272-f001:**
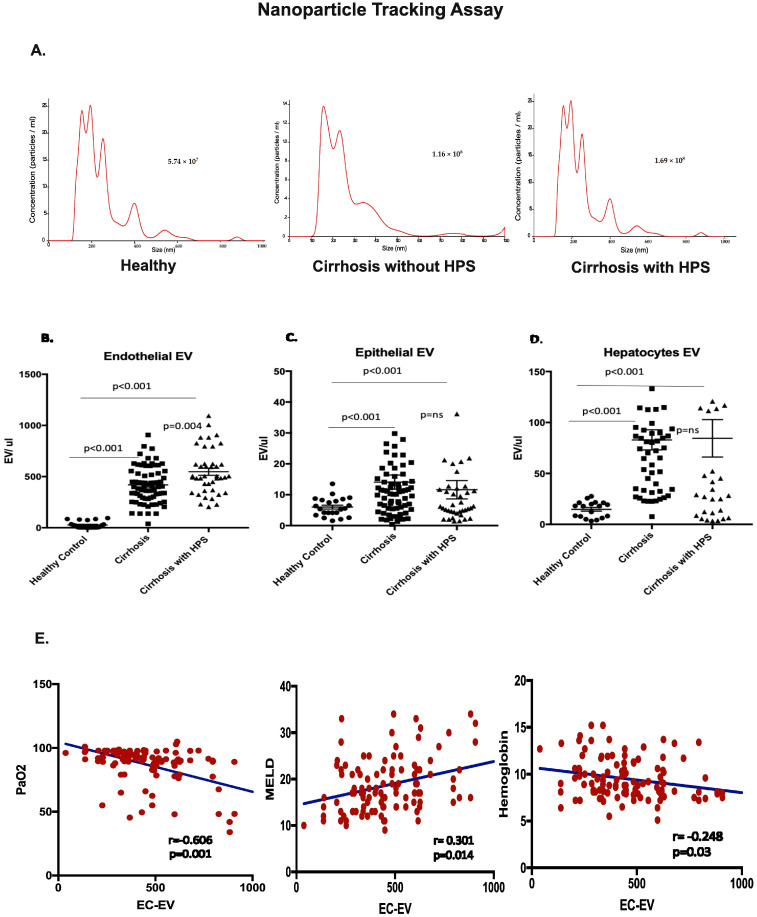
(**A**) Representative images from NanoSight NS300 show the quality, size, and concentration of total EVs isolated from plasma from healthy controls, patients with HPS, and patients without HPS. The Y-axis is concentration (particles/mL), and the X-axis is size in nm. (**B**–**D**) The dot plots show the absolute counts in EV/μL as measured using flow cytometry associated with endothelial cells (EC-EVs), epithelial cell EVs, and hepatocyte EVs. (**E**) The scatter plot shows the EC-EVs correlation with PaO2, MELD, and Hb.

**Figure 2 diagnostics-13-01272-f002:**
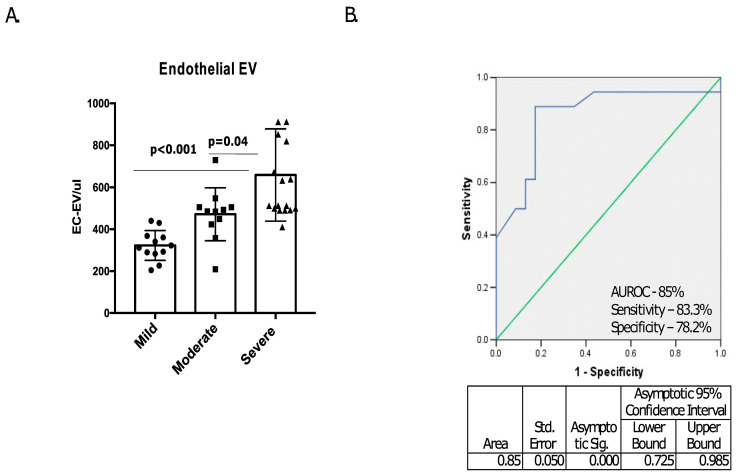
EC-EVs associated with severe intrapulmonary shunting. (**A**) The bar diagram depicts a significant increase in EC-EV levels in severe shunting versus in mild shunting (*p* = 0.00). (**B**) ROC analysis with estimation of the corresponding area under the curve (AUC) shows the predictive potential of EC-EVs for cirrhosis with HPS as mild vs. severe.

**Figure 3 diagnostics-13-01272-f003:**
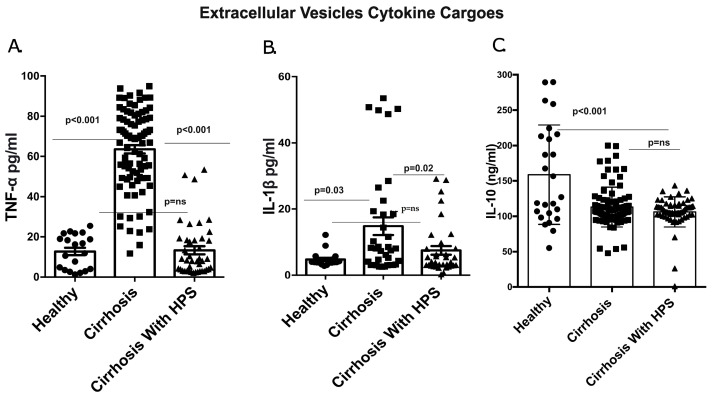
Extracellular vesicle cargoes. (**A**) TNF-α levels inside plasma EVs of cirrhosis patients with and without HPS and healthy controls. (**B**) IL-1 β levels in EVs of cirrhosis patients. (**C**) IL-10 levels in EVs of cirrhosis patients.

**Figure 4 diagnostics-13-01272-f004:**
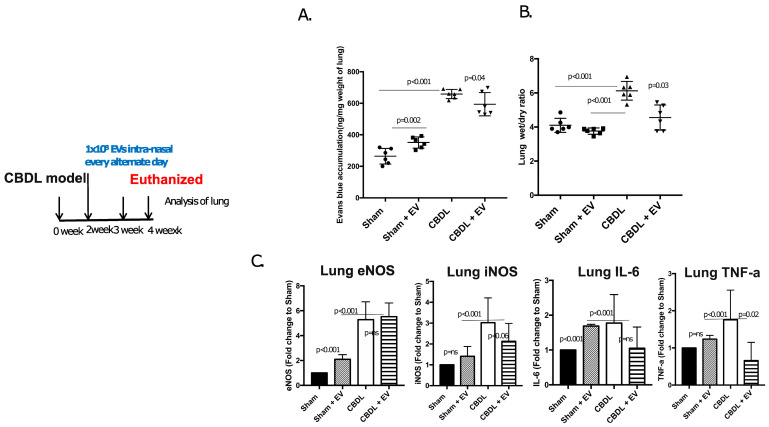
Extracellular vesicles (1 × 10^8^ EV/μL) obtained from healthy WT mice were administered intranasally into CBDL mice (2 weeks post-surgery of CBDL) every day for 2 weeks. (**A**) The bar diagrams show the Evan blue accumulation in lungs and (**B**) liver/body weight in sham-operated and CBDL mice with and without EVs. (**C**) Bar graphs show the fold changes in mRNA candidate genes associated with vasodilators and inflammation.

**Figure 5 diagnostics-13-01272-f005:**
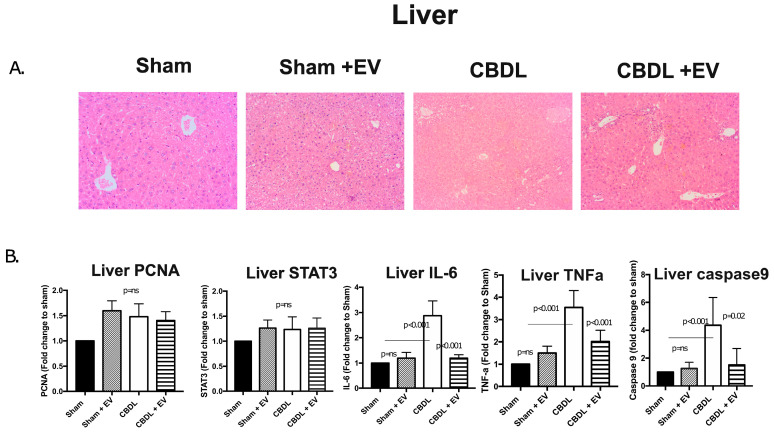
(**A**) Histology depicts ductular proliferation and inflammation, which were improved after intranasal administration of EVs. (**B**) Bar diagram shows the mRNA levels of candidate genes associated with hepatocytes, proliferation, inflammation, and apoptosis.

**Table 1 diagnostics-13-01272-t001:** Clinical characteristics of patients with and without hepatopulmonary syndrome in cirrhosis condition.

S.No.	Variables	Cirrhosis with HPS (*n* = 42)	Cirrhosis without HPS (*n* = 71)	*p*-Value
1	Mean age (SD)	50.6 ± 9.7	50.2 ± 10.7	0.82
2	Gender (males) n (%)	37 (90.2)	67 (94.4)	0.41
3	Mean hemoglobin (g/dL) (SD)	8.6 ± 1.9	10.3 ± 2.0	<0.001
4	Median total leucocyte count (10^3^ cells/mL) (IQR)	5.8 (4.2–8.3)	5.0 (3–7.6)	0.19
5	Median serum sodium (mEq/L) (IQR)	132 (129.3–139.3)	134 (128–136.8)	0.42
6	Median serum potassium (mEq/L) (IQR)	3.9 (3.4–4.4)	3.8 (3.5–4.2)	0.33
7	Median aspartate amino transferase (IU/mL) QR)	58 (39–82)	69 (38–78)	0.93
8	Median alanine amino transferase (IU/mL) (IQR)	30 (18–40)	36 (23–46)	0.11
9	Alcohol n (%)	30 (71%)	19 (86%)	0.75
10	Viral hepatitis n (%)	3 (7%)	1 (4.5%)	0.82
11	Non-alcoholic steatohepatitis n (%)	9 (21%)	2 (9%)	0.08
12	Median Child–Turcott–Pugh (CTP) score	9.1 ± 2.4	8.5 ± 2.2	0.01
13	Median model of end-stage liver disease (MELD) score (IQR)	20.2 (16–25.5)	18.6 (15–22.1)	0.01

## Data Availability

The data that supports the study is available with corresponding author.

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
