# Peer review of "Vascular Extracellular Vesicles Indicate Severe Hepatopulmonary Syndrome in Cirrhosis"

_diagnostics, 2023, doi:10.3390/diagnostics13071272_

Round 1

Reviewer 1 Report

diagnostics-2163366

Abstract
Please define CBDL and EV

Introduction
Define OLT

Methods
Please define ECHO
Line 139: why did the authors choose “injected intranasally”? according to….?

Results
Line 215…erase the phrase “as we know..” and add the correct reference to Kupper cell´s proinflammatory cytokine enhancement. Same situation with lines 216-217

Inform the proinflammatory levels that enhance/decrease among the groups in percentage or another quantitative scale.

Line 233 and the following: why use two weeks old mice? (very young mice), the authors must inform this assay in methodology. According to whom was the timeline decided?

Figure 4, the correct form for “sacrificed” is “euthanized”
In figure 4C, the graphs are too small (enlarged them), and use the same statistics form as in the previous one (line with values)

Figure 5: they lost the  “A” and “B” the authors should change the microphotographs for one that shows the central vein.
Figure 5B: the statistics are lost

Please discuss why the authors did not find differences in IL-10. Why do the authors measure STAT3?  discuss these results. In addition, the authors further discuss the "results",  particularly the relation of EV-inflammation-liver cirrhosis/HPS.

Please send a supplementary with the macro photography of the liver and lung from all murine groups (did the authors measure the hepatic index?).
Also, in the text are a lot of “extra spaces” Please correct it.
LINE 254: “archetype” ??? (no, it isn’t, but it is a  good model)

Author Response

Reply to Reviewers' Comments:

We are very thankful to the reviewer for valuable comments to make the manuscript better. 

Abstract

Please define CBDL and EV

Reply: Common Bile Duct Ligation (CBDL) is an experimental model of obstructive cholestasis induced chronic liver injury wherecommon bile duct is ligated above the duodenum, which is close to human hepatopulmonary syndrome for obstructive cholestasis, liver fibrosis, and portal hypertension in mice. this results in bile duct proliferation, raised liver enzymes,  inflammation and hepatic fibrosis at 3-4 weeks of post-surgery in mouse model.

Extracellular vesicles (EV) are membrane bound cell derived vesicles which are released by all living cells and are present in biological fluid.  

Introduction
Define OLT

Reply: Orthotropic Liver Transplantation (OLT), where source of transplanted liver is from a recently deceased donor.

Methods
Please define ECHO

Reply: Echocardiography is an improved ultrasound imaging diagnostic techniques which provides real-time assessment of intracardiac blood flow with better resolution. This helps HPS patients to diagnose intrapulmonary shunting.

Line 139: why did the authors choose “injected intranasally”? according to….?

Reply: We wanted to target the lungs and the Intranasal is safe, convenient and easy to perform.  Administration of drugs easily cross the nasal mucous membrane reaches  directly to the blood stream through its vascular bed hence provide quick on set of action with 100 % bioavailability as this route avoids gastrointestinal destruction and hepatic metabolism. In view of further translational approach intranasal route was selected.

Results
Line 215…erase the phrase “as we know..” and add the correct reference to Kupper cell´s proinflammatory cytokine enhancement. Same situation with lines 216-217

Reply: We are thankful for the comments. As suggested we have added the reference.

Inform the proinflammatory levels that enhance/decrease among the groups in percentage or another quantitative scale.

Reply: Yes, fold change has been added in the results of cytokines.

Line 233 and the following: why use two weeks old mice? (very young mice), the authors must inform this assay in methodology. According to whom was the timeline decided?

Reply: The administration of EVs were  done  after two weeks of post-surgery of CBDL, however the age of mice were 10-12 weeks, as mentioned in method section on page 3;line 132. We performed the initial experiments and as per the previous reference (CG Tag et.al,2015). In CBDL model the liver injury, fibrosis and inflammation begins by 2 weeks of post-surgery and progresses by week 4 of post-surgery of CBDL with high mortality by week 4-5. 

Figure 4, the correct form for “sacrificed” is “euthanized” 
In figure 4C, the graphs are too small (enlarged them), and use the same statistics form as in the previous one (line with values)

Reply: We again thank the reviewer for critical valuable comments. As suggested we have changed the sacrificed to euthanized. The figures we have tried to attach again with better resolution. In Figure 4C, statistics has been added.

Figure 5: they lost the  “A” and “B” the authors should change the microphotographs for one that shows the central vein.

Reply: We have added the new images of H&E staining of liver in 10x resolution to depict with central vein.
Figure 5B: the statistics are lost

Reply: As suggested we have added the statistics in Figure 5B.

Please discuss why the authors did not find differences in IL-10. Why do the authors measure STAT3?  discuss these results. In addition, the authors further discuss the "results",  particularly the relation of EV-inflammation-liver cirrhosis/HPS.

Reply: IL-10 was analysed as anti-inflammatory cytokine. As suggested has been added in the discussion section.

  1. IL-10 levels in serum has shown to be less in case of cirrhosis than any other liver disease including HCC, viral hepatitis, ACLF and sepsis. Also, as the severity of Liver disease increases a downtrend in IL-10 levels were observed. (Li et al., 2021; Othman et al., 2013)
  2. Moreover, IL10 is an anti-inflammatory cytokines and an immune suppressive and its expression depends upon pro inflammatory signals. (Zhang & Wang, 2006)

IL6/JAK2/STAT3 pathway has shown significance in progression of HPS, high level of STAT3 correlates with increased proliferation of pulmonary microvascular endothelial cell which has been seen as a major underlying mechanism of HPS progression (WANG et al., 2015)

As suggested has been added in the discussion section.

Please send a supplementary with the macro photography of the liver and lung from all murine groups (did the authors measure the hepatic index?).

Reply: We again thank the reviewer for the suggestion. We have added as supplementary figure a macrophotograph of whole mouse with liver and lung. We have not calculated the hepatic index.

Also, in the text are a lot of “extra spaces” Please correct it. 

Reply: We apologise for it. We have tried to remove such errors.

LINE 254: “archetype” ??? (no, it isn’t, but it is a  good model)

Reply: Yes, we agree, we have removed it.

Reviewer 2 Report

Baweja et al. found that EC-EVs from endotelial cells were found in patients with HPS with elevated pro-inflammatory cytokine cargos, indicating severe HPS condition.

The article is well organized. The introduction is based on the current literature. The methodology is appropriate. The authors have used samples from cirrhotic patients with and without HPS, as well as from healthy donors. They have been able to characterize the vesicles obtained from plasma through NTA, western blot with the appropriate markers for EVs, as well as their cargoes via ELISA analyses. They have also developed a mouse model to which the EVs were administered intranasally, being able to demonstrate that it can be a useful model in the study of this hepatic pathology associated with HPS.

I only have a few comments to improve the quality of the article so if authors could consider them, then the article could be published in the journal.

1- Authors should review when Greek symbols are used throughout the document. For example, TNF alpha is sometimes followed by an "a" instead of "alpha".

2-The authors only show one image that they indicate as "representative" of the characterization of the vesicles (fig1A).

Could the authors improve the quality of this image? and/or could they show more images of this type?

3-On the other hand, the authors do not clarify whether there are differences in the size or number of populations present in each of the experimental groups when they have been characterized by NTA

Author Response

Reply to Reviewer’s Comments:

Baweja et al. found that EC-EVs from endothelial cells were found in patients with HPS with elevated pro-inflammatory cytokine cargos, indicating severe HPS condition.

The article is well organized. The introduction is based on the current literature. The methodology is appropriate. The authors have used samples from cirrhotic patients with and without HPS, as well as from healthy donors. They have been able to characterize the vesicles obtained from plasma through NTA, western blot with the appropriate markers for EVs, as well as their cargoes via ELISA analyses. They have also developed a mouse model to which the EVs were administered intranasally, being able to demonstrate that it can be a useful model in the study of this hepatic pathology associated with HPS.

I only have a few comments to improve the quality of the article so if authors could consider them, then the article could be published in the journal.

Reply: We are very thankful for encouragement and valuable comments to make the manuscript better. 

  • Authors should review when Greek symbols are used throughout the document. For example, TNF alpha is sometimes followed by an "a" instead of "alpha".

Reply: We are very thankful for the  comment and agree with your concerns. As suggested, we have changed in the text.

2-The authors only show one image that they indicate as "representative" of the characterization of the vesicles (fig1A).

Could the authors improve the quality of this image? and/or could they show more images of this type?

Reply: We again thank for the help in making the manuscript better. We have added more images of NTA and tried to improve the quality by uploading the images separately.

3-On the other hand, the authors do not clarify whether there are differences in the size or number of populations present in each of the experimental groups when they have been characterized by NTA.

Reply: Thank you for the valuable comment. The levels of EVs from various cell of origin on basis of absolute numbers were calculated using flow cytometry. The NTA was performed to check the quality and concentration only.

Round 2

Reviewer 1 Report

The authors don´t made all the changes solicited. e.g:

-->they should inform the reference related to "intranasal use of EV" and change the word "sacrificed" to "euthanized.

--> Figure 5: they lost the  “A” and “B” the authors should change the microphotographs for one that shows the central vein. ?????

and so others...

Author Response

Reply to Reviewers' Comments:

The authors don´t made all the changes solicited. e.g:

We are very thankful to the reviewer for valuable comments to make the manuscript better.  We extremely apologise to miss out few points.

-->they should inform the reference related to "intranasal use of EV" and change the word "sacrificed" to "euthanized.

Reply: The reference for intranasal use of EV has been added as ref.17 in manuscript. The word sacrificed is changed euthanized and highlighted in the manuscript and figure.

--> Figure 5: they lost the  “A” and “B” the authors should change the microphotographs for one that shows the central vein. ?????

Reply: In the Figure 5, the A and B text had been added. And,  as suggested the microphotographs were changed with showing central vein in the previous version as well.

Old image was:

Changed to New Image now
